# Warming induces unexpectedly high soil respiration in a wet tropical forest

Tana E. Wood [1] ✉, Colin Tucker[2,3], Aura M. Alonso-Rodríguez[4,5], M. Isabel Loza[6], Iana F. Grullón-Penkova [1], Molly A. Cavaleri[7], Christine S. O'Connell[8,9] & Sasha C. Reed[2]

Tropical forests are a dominant regulator of the global carbon cycle, exchanging more carbon dioxide with the atmosphere than any other terrestrial biome. Climate models predict unprecedented climatic warming in tropical regions in the coming decades; however, in situ field warming studies are severely lacking in tropical forests. Here we present results from an in situ warming experiment in Puerto Rico, where soil respiration responses to $^+$4 °C warming were assessed half-hourly for a year. Soil respiration rates were 42-204% higher in warmed relative to ambient plots, representing some of the highest soil respiration rates reported for any terrestrial ecosystem. While respiration rates were significantly higher in the warmed plots, the temperature sensitivity ($Q_{10}$) was 71.7% lower, pointing to a mechanistic shift. Even with reduced $Q_{10}$, if observed soil respiration rates persist in a warmer world, the feedback to future climate could be considerably greater than previously predicted or observed.

Over the next century, air temperatures are forecast to increase between 1 and 6 °C in tropical regions, and tropical soils are expected to warm at a similar rate[1]. This raises the question of how soil respiration in already-warm locales will respond to such change[2–6]. Because of the large magnitude of $CO_2$ exchanged between tropical forest soils and the atmosphere, relatively subtle shifts in soil respiration rates could have a dramatic influence on concentrations of atmospheric $CO_2$ and thus global climate[4–7]. For example, an Earth System Model inter-comparison analysis of the projected carbon (C) balance over the next century showed the tropics to have three times greater variation in model projections relative to any other latitude range, suggesting a poor representation of carbon cycling responses to global change in these carbon-rich ecosystems[5]. The understanding of how temperature regulates soil carbon fluxes in warm tropical forests (mean annual temperatures >20 °C) is particularly poor, in part due to long-held expectations that tropical ecosystems will be less

sensitive to warming than ecosystems in higher latitudes[4,8–14]. However, this paradigm is based primarily on results from laboratory incubation experiments[14–16], elevation gradient studies[17–19], and cross-site comparisons[11], as in situ warming experiments in tropical forests are currently limited to the understory and soil warming experiment described here and a soil-only warming experiment in Panama[5,20–22].

While soil incubation experiments are important methods for exploring mechanisms, their short-term nature, exclusion of roots, and the large disturbance to natural soil structure make it difficult to use them in isolation to ask questions about real-world soil responses to new temperature regimes. In turn, elevational gradient studies cannot capture the responses of tropical biota to continued warming because nowhere on Earth has an analog 'future' climate for lowland tropical forests[9]. Cross-site comparisons are able to take into consideration the long-term adaptation of tropical forests to temperature[11]; however, current changes are happening at a rapid rate and may shift forests to

[1]USDA Forest Service International Institute of Tropical Forestry, Río Piedras, PR, USA. [2]US Geological Survey, Southwest Biological Science Center, Moab, UT, USA. [3]USDA Forest Service Northern Research Station, Houghton, MI, USA. [4]Gund Institute for Environment, University of Vermont, Burlington, VT, USA. [5]Rubenstein School of Environment and Natural Resources, University of Vermont, Burlington, VT, USA. [6]Center for Tree Science, Morton Arboretum, Lisle, IL, USA. [7]College of Forest Resources and Environmental Science, Michigan Technological University, Houghton, MI, USA. [8]Department of Environmental Studies, Macalester College, St. Paul, Minnesota, USA. [9]Biology Program, Schmid College of Science and Technology, Chapman University, Orange, CA, USA. ✉ e-mail: tana.e.wood@usda.gov

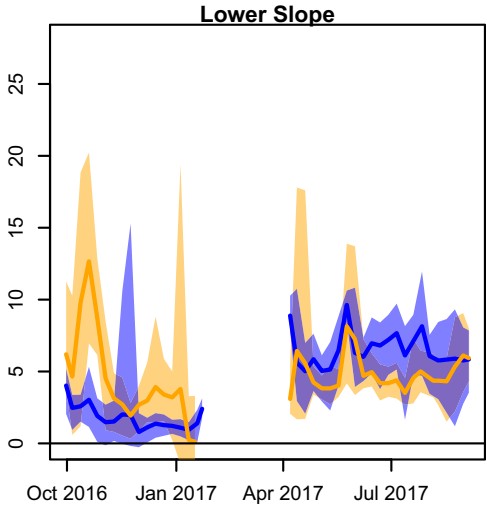

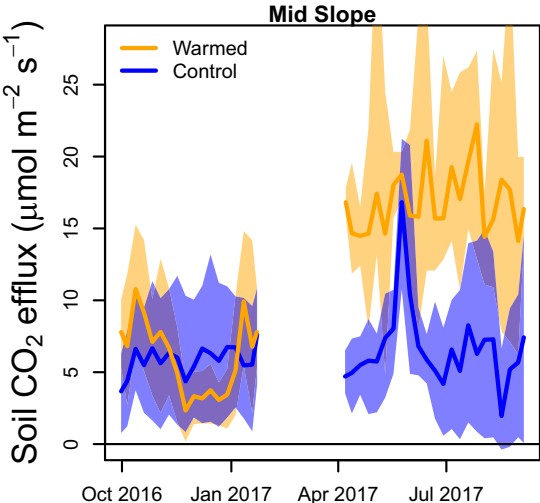

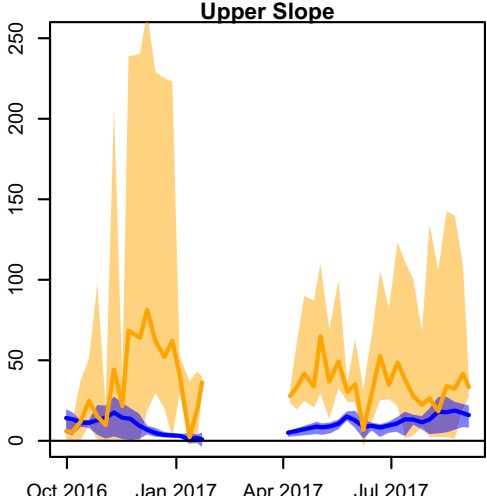

**Fig. 1 | Soil respiration rates in warmed and control plots.** Weekly mean soil $CO_2$ flux for control (blue line) and warmed (orange line) plots with the 95 % confidence intervals (shaded) for the 1-year duration of the study (September 2016–2017). The top panel is the average soil respiration rates for the Lower slope paired plots, the middle panel is average rates for the Mid slope paired plots, and the bottom panel is average rates for the Upper slope paired plots. The break in data occurred due to maintenance being provided to the automated soil respiration equipment. Note that the y-axes have different scales due to the differences in rates.

evaluate soil respiration responses to chronic warming[20]. We experimentally warmed understory plants and soils 4 °C above ambient temperatures to a depth of at least 50 cm using a hexagonal array of infrared heaters. Three warmed plots were paired with three ambient (control) plots of similar topographic position (Lower slope, Mid slope, and Upper slope). For a year, half-hourly soil respiration rates were measured in each plot using automated soil respiration chambers, resulting in 57,450 rate measurements for our study period of September 28, 2016, to September 5, 2017. In addition to the high temporal resolution soil respiration fluxes measured within the temperature manipulation plots, we performed an additional 152 flux measurements between November 4–19, 2020, across the larger field site to assess spatial heterogeneity in soil respiration rates.

## Results and discussion

Experimental warming resulted in substantial, larger than expected increases in soil respiration in plots that were experimentally warmed relative to the unwarmed control plots (Figs. 1 and 2). Warming increased $CO_2$ flux by 42% and 59% in Mid and Lower slope positions, respectively (paired $T$-test, df = 24,521 pairs [Lower slope]; df = 15,635 pairs [Mid slope]). In addition to warming-induced increases in soil respiration means, warming increased the variability of the rates, which may represent an increased variability in environmental conditions or a synergy between warming and soil moisture controls. The positive response of $CO_2$ flux to in situ warming was substantially higher in the Upper slope position relative to the lower topographic positions (204%; df = 12,910 pairs). For two of the paired plots (Lower and Mid slope), the warming-induced increases in respiration rates were in the upper range of what has been reported for higher latitude forests; however, the relative responses of the Upper slope were greater than rates observed in any field warming experiment, regardless of ecosystem type or methodology[7]. Given the already high soil respiration fluxes characteristic of this and other tropical forests[7], warming resulted in a large amount of additional $CO_2$ released to the atmosphere, with 6.5, 9.7, and 81.7 Mg $CO_2$-C ha[-1] yr[-1] more $CO_2$ being released in the warmed plots compared to the control plots in the Lower, Mid, and Upper slope positions, respectively. Specifically, the additional soil $CO_2$ released from the Mid slope was equivalent to the total annual net primary productivity (NPP) of a temperate grassland, and the Lower slope additional $CO_2$ released was equivalent to the total annual NPP of a temperate deciduous forest. For the Upper slope, there is nothing on record that compares to the additional $CO_2$ flux other than the conversion of a tropical peatland forest to an oil palm plantation[7,23].

The inclusion of warmed understory plant communities combined with warmer soils in this in situ warming experiment allows for an integrated exploration of above versus belowground controls on soil respiration responses. This inclusion is important because plants grown under warmer temperatures could alter carbon allocation to root biomass and/or root exudates that, in turn, may influence the contribution of root respiration to total soil respiration, as well as the available carbon and fuel for microbial processes[24]. Acclimation of root-specific respiration under warming conditions could also shift the contribution of roots to total soil respiration[25]. In addition, changes to the microbial community, microbial activity, changes to abiotic or geochemical conditions, or some combination may drive observed

new regimes at a pace at which they may be unable to adjust[10]. Further, while in situ field warming experiments that warm only the soil provide invaluable insights, they are limited in their ability to interpret how changing plant ecophysiology may interact with changes to belowground processes. In response to these limitations, we developed an in situ field warming experiment – Tropical Responses to Altered Climate Experiment [TRACE] – in a tropical forested ecosystem to

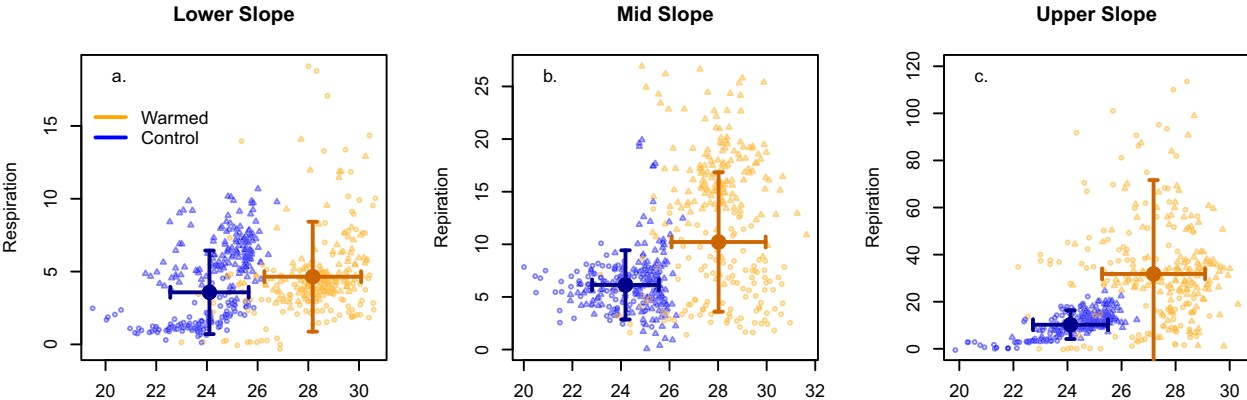

**Fig. 2 | Diurnal mean temperature versus diurnal mean respiration rate by topographic position and treatment.** Control (black) and warmed (red) observations across the study period in the **a** Lower, **b** Mid, and **c** Upper slope paired plots. Error bars indicate standard deviation. See Supplementary Fig. S5 for visualization of this figure with the same scale y-axis.

increases in soil respiration rates in response to warming[26,27]. Understanding how plant and soil responses to warming individually and interactively affect soil respiration is critical for accurately modeling and forecasting feedbacks to carbon cycling and future climate. While spatially variable, live fine root biomass did not differ among warmed and control plots prior to starting the warming treatment; however, after just six months of warming, live fine root biomass was 32% lower in warmed relative to control plots (Supplementary Fig. S1; two-way ANOVA $P = 0.04$, $F = 5.83$; followed by Tukey's test $P < 0.05$). Further, live fine root biomass of the Upper slope plots did not differ significantly from live fine root biomass in the Lower or Mid slope topographic positions (two-way ANOVA $P > 0.05$), suggesting cross-slope patterns were not due to topographic differences in root biomass. We observed no significant acclimation of root-specific respiration in response to warming, but, when combined with the reduced root biomass, the contribution of roots to total respiration declined[28]. Although total and extractable carbon pools weren't altered by warming, soil microbial biomass carbon did increase significantly, growing by over 50% in the warmed plots relative to the unwarmed control plots[29].

Taken together, these results suggest that the large increase in soil respiration rates in response to warming was due to an increase in microbial-derived $CO_2$ efflux, while simultaneously experiencing a concomitant decline in the contribution of root-derived $CO_2$. Similarly, a belowground-only in situ soil warming experiment in Panama found a large increase in soil respiration in response to warming, with increases in $CO_2$ efflux that were primarily derived from microbial sources with no change in root contribution[22]. It is possible that the reduced allocation to roots by plants observed at our site was driven by warming of the aboveground vegetation, and not by warmer soils. These differential results highlight the potential for above- and belowground interactions to play an important role when considering the key drivers of the response of soil respiration fluxes to warmer temperatures in tropical ecosystems.

Given the especially high rates of soil $CO_2$ efflux, we used several approaches to confirm that our observed emissions increases were indeed primarily driven by warming and not by stochastic spatial variability or measurement error (Supplementary Materials and "Methods" section). Whereas the Upper slope is clearly a "hotspot" for higher soil respiration rates, throughout the study, all plots exhibited extreme "hot moments" of soil respiration, where observed flux values were several times higher than the mean value (Fig. 3), which is characteristic of trace gas fluxes in this system[30]. While there was high temporal variation throughout the year, we found no significant diurnal variation in soil respiration. We further conducted a spatial survey of 30 locations randomly selected outside of the plot locations

across the 40 m × 60 m TRACE area to quantify the spatial variability of soil $CO_2$ fluxes across the TRACE landscape (Supplementary Fig. S2). We unsurprisingly found variability across space but no evidence that the warmed plots were systematically located on landscape-level hot spots. Additionally, pre-treatment data showed no statistical differences in soil respiration rates between paired warmed and unwarmed plots prior to the initiation of warming (Supplementary Fig. S3), and running a generalized least squares model with and then without the Upper slope (i.e., the control-warming paired plots with particularly large differences in $CO_2$ flux rates) did not change the finding that warmed plots had significantly higher soil respiration rates (Supplementary Table S1, Supplementary Materials and "Methods" section). Field sensitivity analyses failed to detect evidence that invertebrate or animal presence in the chamber could have produced such high respiration rates (Supplementary Materials and "Methods" section).

Despite sustained high rates of soil $CO_2$ flux, we found no significant difference in total or available (i.e., extractable) carbon in the surface soils (0–10 cm) of the plots either before or after 6 months of warming[29]. It remains unclear whether or for how long soil carbon stocks can be sustained at current levels despite such high $CO_2$ flux rates, and it will be critical to determine the source of additional carbon that is fueling elevated soil respiration rates, particularly in the Upper Slope. For example, there could be greater carbon loss from deeper soils[31], which could be especially relevant for the Upper slope, where the soils are considerably deeper[32]. Inputs from the declining fine root pool, altered litter layer decomposition, or a concentration of preferential flow paths in the more aerated upper slopes could also explain the high fluxes[33]. There could also be abiotic factors influencing soil carbon availability via an increase in desorption reactions or a shift in the physicochemical environment[26,34]. Ascertaining which of these potential sources of carbon are driving our responses requires further study. That said, regardless of the carbon source, our observed increases in soil respiration rates in response to warming are conspicuously greater than the 95% range of that observed for warming experiments in northern hemisphere forests (12–31% compared to our observed 42–204%)[35], suggesting large temperature sensitivity of the tropical forest carbon cycle. Further, there were notable increases in the variability of soil respiration at higher temperatures (Fig. 1). Increased variability in soil $CO_2$ flux under chronically warmer conditions has substantial implications for the equilibrium soil C stock size if highly variable emissions are not counteracted by similarly variable and large soil C inputs. These responses in both the mean and variation provide critical insight into quantifying tropical forest feedbacks to climate change in a range of conceptual and numerical models, and also contradict the supposition that tropical forests may be relatively insensitive to warming. In addition, our observed increase in microbial

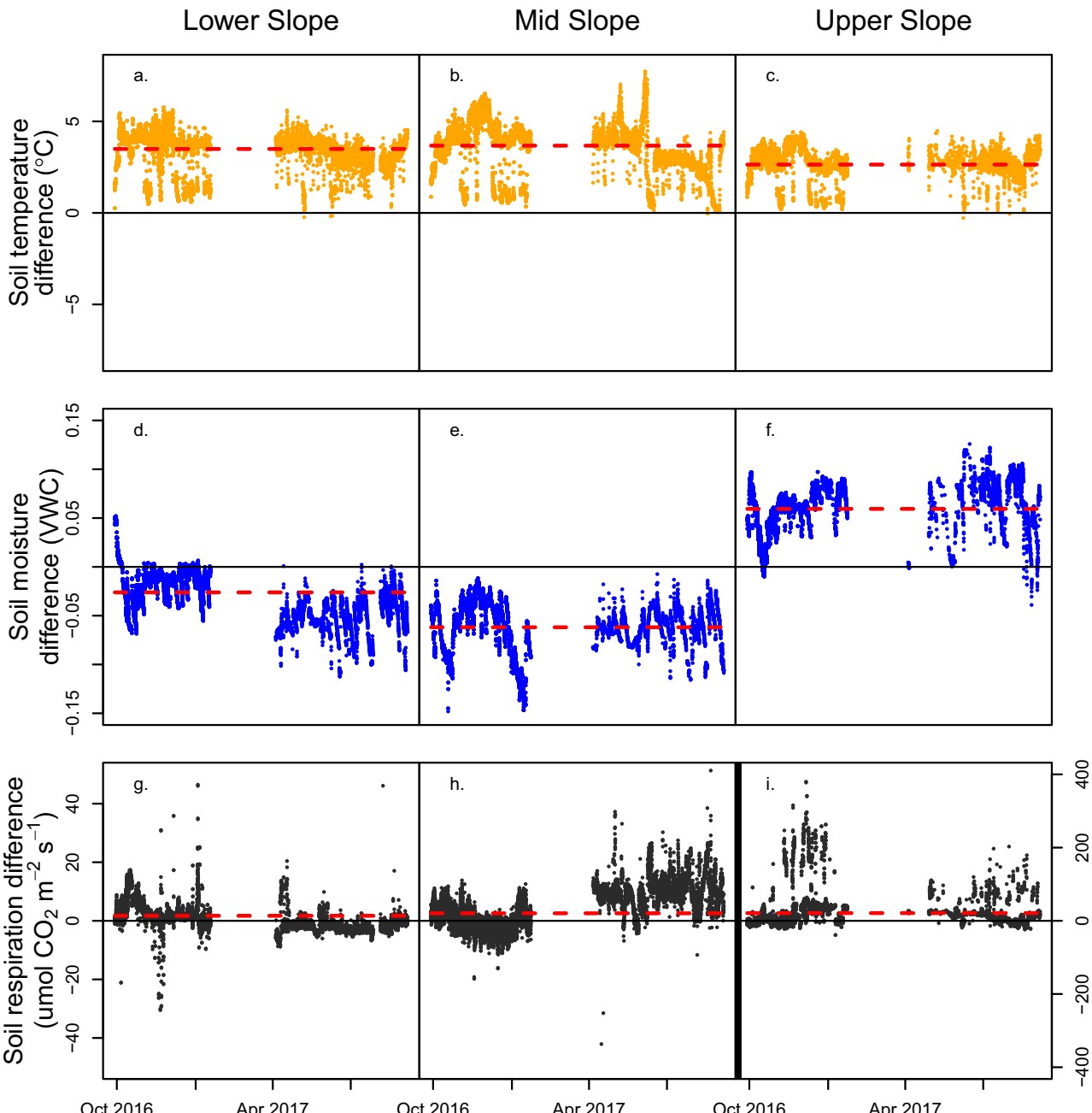

**Fig. 3 | Half-hourly differences between soil temperature, volumetric water content, and soil respiration over the course of the 1-year study period.** Half-hourly differences between soil temperature, volumetric water content, and soil respiration over the course of the 1-year study period. The top row (**a**–**c**) shows the difference in soil temperature in °C for Lower Slope (**a**), Mid Slope (**b**), and Upper Slope (**c**) paired plots. The middle row (**d**–**f**) shows the difference in soil moisture in volumetric water content (VWC) for Lower Slope (**d**), Mid Slope (**e**), and Upper Slope (**f**) paired plots. The bottom row (**g**–**i**) shows the difference in soil respiration rate (umol $CO_2$ m$^{-2}$ s$^{-1}$) for Lower Slope (**g**), Mid Slope (**h**), and Upper Slope (**i**) paired plots. Note the order of magnitude difference in scale between the Upper slope location (scale to the right) and the Lower and Mid slope locations (both scales to the left) for soil respiration values (separated by a doubled horizontal line between panels h and i). The horizontal dashed red lines show the mean difference (warmed minus control) between paired warmed and control plots across all measurements. The significance of the ANOVA was $p < 0.001$ for all comparisons.

biomass carbon (-50%) was greater than the increase observed in the soil warming experiment in Panama or any northern hemisphere ecosystem[22], and is contrary to observations from tropical elevation studies[19], highlighting the exceptional value of additional data that elucidate in situ tropical forest responses to warming in global assessments[19,22,35].

Because temperature and moisture often co-vary, we also explored how the application of the warming treatment affected both soil temperature and moisture at the site. Soil temperatures increased similarly across the warmed plots at all topographic positions in response to experimental warming and were on average 3.99 °C warmer than control plots (Fig. 3a–c). In contrast, while soil volumetric water content (VWC) did not differ among the plots prior to starting the treatment[20], soil moisture responded very differently to warming depending on topographic position (Fig. 3d–f). Soils in the Lower and Mid slope plots were drier under the warming treatment (−0.03 g $H_2O$ g$^{-1}$ soil [6.8% lower] for Lower slope and −0.06 g $H_2O$ g$^{-1}$ soil [16% lower] for Mid slope between warmed and control plots, $p < 0.001$),

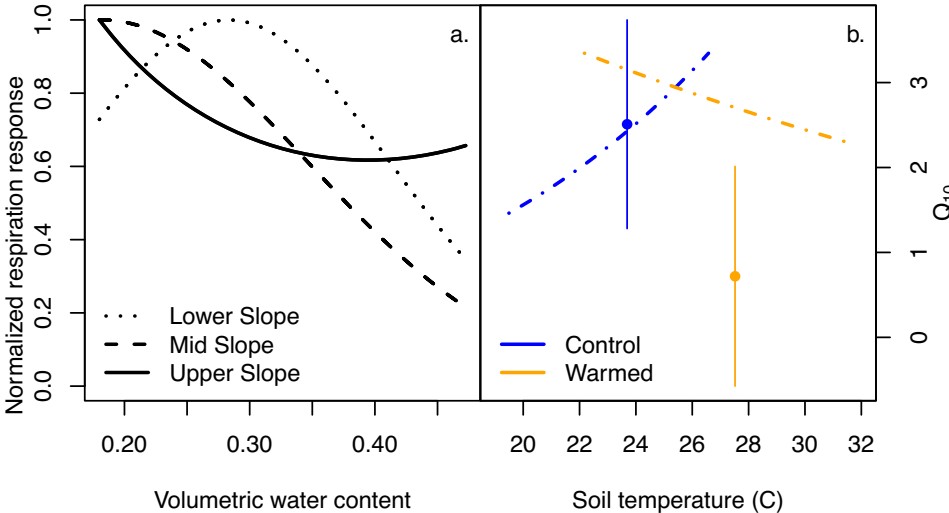

**Fig. 4 | Volumetric water content (VWC) and temperature response curves from generalized least squares models. a** Fitted curves showing the normalized respiration response (0–1) to soil volumetric water content, with slope position identified with different line patterns. **b** Fitted curves for control (blue) and warmed (orange) plots showing the normalized respiration response (0–1) to soil temperature, with data on the y-axis shown in **a**). Also on (**b**), the right-hand y-axis shows the $Q_{10}$ temperature sensitivity (filled circles ± s.e.) for control (blue) and warmed (orange), with the $Q_{10}$ value positioned at the mean temperature for the control and warmed treatments.

while the soils of the Upper slope were surprisingly wetter in the warmed plots compared to control (+0.06 g $H_2O$ g$^{-1}$ soil [18% higher], $p < 0.001$, Fig. 3). There is a range of explanations for the occurrence of greater soil moisture in the warmed Upper slope plot when compared with the control, including lower litter quality combined with extensive drying of the litter layer, which could slow decomposition and create a buffer against soil moisture loss[36,37]. More work is needed to isolate the cause. Regardless, these data suggest that warming will, in part, regulate soil respiration through interactions with soil moisture, and that in situ experiments could further elucidate interactive controls with longer-term assessments of the connections among soil temperature, moisture, and biological activity.

Within the context of a warming climate, it is worth noting that lowland tropical forests experience a very narrow temperature range compared to all other terrestrial environments on Earth. In this system, a mean experimental increase of 4 °C nearly doubled the total ecosystem temperature range. Specifically, the mean diurnal temperatures of the control plots (-20–26 °C) overlap less than half of the range of the warmed plots (-22–32 °C, Figs. 2 and 4). Therefore, while the magnitude of warming that these soils experienced was in line with other ecosystem warming studies[38], the temperature increase relative to the temperature range was well beyond that imposed in other aboveground plus belowground field warming studies and pushed the warmed plots into a new climate space. The future climate warming expected in tropical forests[1,4,8–10,] will also have these large proportional increases in temperature.

Using diurnal averaged soil respiration rates, temperature, water content, and topographic position, we evaluated soil temperature and soil moisture effects on soil respiration using a generalized least squares modeling approach[39] (Supplementary Materials and "Methods" section). Model results revealed that, regardless of warming treatment, the relationship between normalized respiration response and soil moisture on Lower and Mid slope positions was significantly negative ($p < 0.001$, Supplementary Table S1, Supplementary Fig. S4), while Upper slope positions, while also having a significantly negative relationship between soil respiration rate and moisture, exhibited a significantly positive interaction term between normalized respiration response and soil moisture ($p < 0.05$, Supplementary Table S1, Supplementary Fig. S4). In other words, soil respiration in the lower

topographic positions declined during times when soils were wetter (i.e., after rainfall), consistent with decreased soil aeration[40], but this relationship was much weaker in Upper slope positions. The change in this statistical relationship was driven by occasional extreme respiration rates, which were observed in Upper slope plots during periods of high soil moisture values, regardless of warming status (Figs. 3 and 4). This points to more complex moisture and temperature interactions than expected for this system.

Over time, there is potential for organisms to adapt or acclimate to their new environment[41]. Thus, we derived the $Q_{10}$ of soil respiration for each of the plots (defined as the multiplicative change in soil respiration for every 10 °C increase in temperature, Fig. 4b, Supplementary Materials, and "Methods" section) to explore if organisms in the warmed plots had an altered relationship between warming and respiration rates. $Q_{10}$ was significantly reduced with warming from a $Q_{10}$ of 2.51 ± 1.23 (controls) to a $Q_{10}$ of 0.71 ± 1.30 (warmed; Fig. 4b, $p < 0.001$). The $Q_{10}$ values of the control plots fell within the range of the global mean of 1.6–3[7,42–44], while the mean $Q_{10}$ of the warmed plots falls well below that range. These $Q_{10}$ values indicate that soil respiration rate decreased per unit of increased temperature in the warmed plots (i.e., $Q_{10} < 1$), despite the fact that respiration rates themselves were higher in the warmed plots. In summary, while the sensitivity to temperature was reduced in response to warming, the higher respiration rates suggest a shift toward overall higher basal metabolic rates. Crucially, this finding suggests that the microbial community is exhibiting a highly plastic response to chronic +4 °C warming, with acclimatization or adaptation responses that both compensate for (reduced $Q_{10}$) and enhance (higher respiration rates) the response to temperature[14]. This has important implications not only for the relationship between temperature and $CO_2$ efflux for such carbon-rich forests, but also for the use of a single $Q_{10}$ in Earth System Model forecasts of future climate.

Overall, the higher respiration rates in the warmed relative to control plots, independent of temperature sensitivity, coincided with a significantly higher soil microbial biomass in the warmed plots, which has been shown to correlate with higher soil respiration rates[14]. While soil respiration in the warmed plots was higher across all temperatures (Figs. 2 and 3), the sensitivity (i.e., slope) of soil respiration in the warmed plots was flat to negative (Fig. 4b), suggesting that warming

induced a systemic shift in function. Given the decline in root biomass observed in the warmed plots, we attribute this shift in function to heterotrophic rather than autotrophic responses. These functional shifts could include a change in microbial carbon use efficiency, a shift in the microbial community composition[17,45], and/or a change in the distribution of biotic activity vertically through the soil profile[31] (e.g., driven by warming-induced changes to soil hydrology).

Contrary to the long-held paradigm that tropical forest responses to increased temperature will be relatively muted in their role in climate change feedbacks[11,13], we found large increases in soil respiration rates in response to in situ experimental warming that interacted in complex ways with soil moisture. Due to the amount of carbon released, these increased rates have substantial implications for forecasts of future climate at the global scale. While soil respiration showed signs of acclimation with respect to $Q_{10}$, the respiration rates were substantially higher in warmed relative to control plots (+42–205%) across all three topographic positions. This study contributes the second field-based observation from an in situ experiment that a warming climate may lead to large increases in $CO_2$ fluxes from carbon-rich tropical soils, adding observations from a a-seasonal wet tropical forest on highly weathered, deep soils to the previous results from a seasonally dry tropical forest with relatively shallow soils[22]. Taken together, the work demonstrates a potential for large carbon losses from tropical forest ecosystems in a warmer world, and highlights the immense value in evaluating the response of soil respiration to warmer temperatures across a range of tropical forested ecosystems, which include 30 Holdridge Life Zones spanning lowland dry deciduous to montane wet evergreen[46,47]. Each of these forest types reflects an incredible level of diversity in forest structure, community composition, and soil conditions that make it unlikely these systems will exhibit a single response trajectory to a changing climate[7]. While experimental warming resulted in a substantial increase in soil respiration at both this forest site and the site in Panama, the underlying controls, as well as the magnitude of the response, appear to be different. For example, our data suggested some acclimation/adaptation of respiration rates combined with a decrease in root respiration contributions to $CO_2$ flux, whereas the seasonally dry forest in Panama found no significant effect of warming on root-derived $CO_2$ flux and no signs of adaptation/acclimation. At both sites, large amounts of additional carbon continued to be released to the atmosphere from warmed plots with no signs of declining in the first 1–2 years after initiating warming, though how this will moderate with time remains to be seen. Nevertheless, understanding the underlying mechanisms driving the response of soil respiration to warmer temperatures is critical for accurate representation of tropical ecosystems in global models and assessing the magnitude and duration of feedback to future climate over the long term.

## Methods

### Site

The TRACE experiment is located in the Luquillo Experimental Forest in northeastern Puerto Rico (18.32465° N, 65.73058° W, 100 m a.s.l., 24 °C mean annual temperature, 3500 mm mean annual precipitation)[20]. Three open-air replicate 12 m² hexagonal plots were warmed with an array of six infrared heaters in each plot, increasing soil temperatures by +4 °C above average ambient. Three control plots were created with the same dimensions and infrastructure, but with no warming[9]. Plots were paired by topographic position within the site (Lower slope, Mid slope, Upper slope). The warming treatment began September 28, 2016, using a feedback control system that acts concurrently and independently at the plot scale to maintain a fixed temperature above the mean ambient temperature of all control plots[20]. Each plot was instrumented with Campbell CS655 soil moisture and temperature probes at three depths (10 cm, 30 cm,

50 cm; Campbell Scientific, Logan, Utah, USA). On average, we achieved 24-h per day soil warming of 3.69–4.02 °C above ambient at 10 cm depth with significant warming to 50 cm. The warming treatment was stopped on September 5, 2017, when Hurricane Irma and then Maria passed over the island of Puerto Rico, and power to the experiment was lost.

### Soil respiration

Soil respiration was measured with a soil $CO_2$ flux analyzer control unit (LI-8100A) connected to a multiplexer (LI-8150) and six Long-Term Chambers (8100-104; LiCor Biological Sciences; Nebraska, USA). A permanent PVC soil collar (21.34 cm o.d.) was inserted 5 cm near the center of each plot one month prior to starting measurements. Prior to the start of the warming treatment, several long-term chambers were under repair, and thus, we collected soil respiration measurements from each plot in field campaigns such that three continuous measurements were collected per day per plot for a total of nine days. This survey was conducted with the same long-term chamber and hose/cable extension assembly as used in the automated measurements, as well as the same programming configuration used during the long-term automated measurements, changing the chamber offsets before each measurement to account for the repositioning of the chamber. The field campaigns provided us with a reliable baseline comparison of all the plots prior to the warming treatment, with no pre-treatment differences in soil respiration rates between warmed and control plots during this period (Supplementary Fig. S3). On the day that the warming treatment began (September 23, 2016), one long-term chamber was installed in all six plots, and half-hourly respiration measurements were begun. Equipment ran almost continuously except for a period in November when all six long-term chambers were removed for maintenance. Flux validation efforts included both computational and field checks (Supplementary Materials and "Methods" section).

A further 152 flux measurements were taken in fall 2020 (from November 4 to 19, 2020) across the full TRACE field site to assess spatial heterogeneity in soil $CO_2$ flux rates and characterize whether the rates seen in the six experimental TRACE plots were representative of the overall spatial heterogeneity. We collected $CO_2$ measurements from 30 collars installed outside the TRACE plots and from the 6 collars installed previously in each experimental plot. We took 4 measurements from most collars located outside of the plots, 5 measurements from the collars inside of the plots, and 3 observations from 2 of the collars outside of the plots due to inclement weather during field work. Spatial interpolation was used to create a flux spatial variability map (Supplementary Fig. S2).

### Root biomass and soil carbon and nutrient analyses

Three soil cores of 10 cm depth and 5 cm diameter were collected from each of the six plots in March 2016 (prior to warming), March 2017 (6 months after warming), and September 2017 (one year after warming). Within 24 h of collection, fine roots (<2 mm diameter) were hand-sorted from cores and subdivided into live and dead. Roots were then washed and dried to obtain total root biomass for each of the cores. Root biomass data were not obtained from the September 2017 cores due to the imminent arrival of Hurricane Maria; however, the complete suite of biogeochemical analyses was performed.

After roots were removed, total soil carbon concentrations were assessed on oven-dried (60 °C for 48 h) soils using an elemental analyzer (Elementar Americas, Mt. Laurel, NJ, USA). Extractable carbon was assessed by shaking fresh soils (on the same day of collection) with 0.5 M $K_2SO_4$ for one hour, and filtering using Whatman GF/F glass fiber filter paper (Whatman International, Springfield Mill, UK). Extracts were analyzed using a Shimadzu TOC-Vcpn/TN-1 (Shimadzu Corporation, Kyoto, Japan).

## Statistical analyses

To determine the effect of infrared warming on soil temperature (10 cm depth), moisture (VWC; 10 cm depth), and respiration, measurements of each variable from paired warmed and control plots were paired in time, and then the difference between the respective variables in warmed and control plots at each time step was calculated. We then used a one-sample $T$-test in R version 3.5.3 ("Great Truth", R Core Team 2019) to evaluate whether the differences were >0. The application of the $T$-test to these data was based on the following: the very large sample size permitted some deviation from the assumption of normality of small-sample $T$-tests, based on the central limit theorem[48]. It is important to note that while measurements of temperature and moisture within each time series are not independent, the differences between the paired time series can indeed be considered independent. We compared data from before (March 2016) and after warming (March 2017) with a two-way ANOVA (Treatment*Time), using the mean biomass of three replicate cores per plot to assess treatment effects on root biomass. To evaluate whether topographic position affected live fine root biomass, a second two-way ANOVA (Treatment*Slope) was performed, using the March 2017 data of all replicates per plot to obtain enough replication per slope (upper, mid, lower). For both analyses, Tukey's *post-hoc* test was used for pair-wise comparisons within groups of significant main effects. All analyses were conducted with the aov and TukeyHSD functions of the stats package in R version 3.6.0 ("Planting of a Tree").

## Modeling

We modeled respiration as a function of temperature and moisture using a mixed modeling approach as follows. Using diurnal averaged soil respiration, temperature, and water content, we evaluated treatment effects on soil respiration using a mixed effects model implemented in the 'nlme' package in R[39] in R version 3.6.0 ("Planting of a Tree"). We log-transformed the respiration rate and used the 2nd-order polynomial fit of soil moisture, based on a priori expectations of the temperature and moisture responses of soil respiration. We evaluated both linear mixed effects, with chamber random effects, and a generalized least squares model with different variance structures and approaches to temporal autocorrelation within a chamber. Model selection was conducted using Bayesian Information Criteria (BIC) to determine the variance and autocorrelation structure for repeated measurements within a chamber across time. After the best variance structure was found, backward selection was applied (using BIC) to determine the best set of fixed effects to include in the model. Parameter estimates for fixed effects (including warming treatment level, soil moisture, soil temperature, and hillslope location) were based on the subset of parameters and interactions that were significant in the best-fit model. The best model was fitted via generalized least squares and accounted for autocorrelation of fluxes across days within a chamber as a first-order autoregressive process with an estimated phi = 0.78, where variance was estimated as a function of measurement temperature and was allowed to vary between chambers[49]. The best model excluded interactions between soil temperature and hillslope location, as well as the interaction between warming treatment and soil moisture, but retained the main effects of each of these, as well as the interaction between hillslope location and vwc, and warming treatment and soil temperature. $Q_{10}$ values were estimated by applying the transformation $Q_{10} = \exp(10 * \beta)$ to the temperature response parameters ($\beta$) in the GLS analysis. The model is described in detail in the Supplementary Information (Supplementary Materials and "Methods" section).

## Data availability

The data generated in this study have been deposited in the Environmental System Science Data Infrastructure for a Virtual Ecosystem (ESS-DIVE) Database [https://doi.org/10.15485/2568416].

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

## Acknowledgements

This research was supported by the US Department of Energy, Office of Science, Office of Biological and Environmental Research (awards DE-SC0012000, DE-SC-0011806, 89243018S-SC-000014, 89243018S-SC-000017, DE-SC-0018942, DE-SC0022095, DE-SC0025314, and 89243021S-SC-000076); the National Science Foundation (award DEB-1754713); USDA Forest Service International Institute of Tropical Forestry and the University of Puerto Rico-Rio Piedras. We are grateful to R. Reibold and A. Howell for their substantial contributions to data collection and to M. Berberich and D. Iglesias-Ortiz for their contribution to quality assessment and control of the data. Assistance with experimental operation and data collection was provided by numerous volunteers and summer interns. We thank M. Keller, A. Walker, R. Norby, and A. Lugo for helpful comments on the manuscript. Any use of trade, firm, or product names is for descriptive purposes only and does not imply endorsement by the U.S. Government.

## Author contributions

T.E.W., M.A.C., and S.C.R. designed the overall experiment. T.E.W. and A.M.A.R. designed the study reported herein, and T.E.W., A.M.A.R., M.I.L., and I.G.P. coordinated the day-to-day field measurements. A.M.A.R., M.I.L., and I.G.P. processed the data. M.I.L. and I.G.P. conducted the spatial survey. C.T. and C.S.O. analyzed the data and created the figures. T.E.W. wrote the first draft and jointly wrote subsequent drafts of the manuscript with the other co-authors.

## Competing interests

The authors declare no competing interests.
