## [Peer Review file · Nature Communications]

Warming induces unexpectedly high soil respiration in a wet tropical forest

Corresponding Author: Dr Tana Wood

Version 0:

Reviewer comments:

Reviewer #1

(Remarks to the Author)

Tana Wood and co-authors present data from a pioneering soil warming study in the tropics. To my knowledge this is the first manipulative warming study in the tropics and the results are of high interest since modelling exercises suggested that warming effects on soil and ecosystem carbon cycling could be especially responsive in these (warm-wet) ecosystems. Authors basically present a one year time series from six auto-chambers measuring soil CO₂ efflux and observed a strong positive response of soil CO₂ efflux to artificial warming. Though I think this study gives some strong indications, I am not convinced that the broad conclusions are justified by the dataset. The simple reason is the inefficient spatial replication of soil CO₂ efflux measurements.

Specific comments:

Soil CO₂ efflux typically shows high spatial variation. In the presented study, soil CO₂ efflux was measured at totally 6 spots (3 chambers at control plots, 3 chambers at warmed plots; n =3). On top, chambers (plots) were aggregated along slope positions (lower slope, middle slope, upper slope; n=1) with regard to statistics and the interpretation of the results. Accordingly this is the lowest amount of spatial replications possible for doing any statistics but certainly does not cover much spatial heterogeneity in fluxes. Authors suggest that the individual auto-chambers produced high temporal resolution measurements (several thousand flux estimates per year) and that the paired-in-time-differences between the treatment and control chambers can be used as kind of time in space substitution. This is not so convincing to me. Even with some thousand temporal replicates of a single chamber one cannot exclude that (I) any chamber showed some temporal malfunctions and (II) that unexpected other sources temporarily contributed to the CO₂ production in single chambers. The extremely high CO₂ efflux at the warmed upslope chamber might be such an example. The high emissions are difficult to explain except that there was some chamber malfunction (indicator would be that there were only half as much pairs upslope as mid- and downslopes, L59-62, Fig2) or that some insects, mammals, or whatever, temporarily occupied the chamber and contributed to the CO₂ production (indicator for that would be the rather erratic spikes in CO₂ differences of up to 400,mmol in fig 2 for the upper slope position – such spikes can hardly be explained with enzyme kinetics of decomposer microbes). Well, such a kind of outlayer-chamber is not really a problem if it is only one of many spatial replicates. It, however, becomes a real issue if there is only one chamber available, such as it was the case here.

There is some general contradiction in the argumentation. It is suggested that the temperature sensitivity of the soil CO₂ efflux decreases with the 4°C warming (from about 2 to about 1). How can then a 4°C warming lead to a 200% increase in soil CO₂ efflux? The Q10 would be far beyond.

It is argued that the high temporal resolution allowed covering “hot moments” in soil respiration. Indeed there were fluxes up to 400µmol but can such fluxes be explained with any logical enzyme kinetics or whatever related to SOM decomposition response the 4°C increase in soil temperature? Q10 extrapolated from seasonal data always needs to be taken with caution as many other factors aside soil temperature affect the temporal course of soil CO₂ efflux.

If warming really induced a loss of 96 Mg C ha⁻¹ y⁻¹ at the upslope site, then the whole soil C (numbers are not reported in the paper) of the site might be respired within a few years. Is this reasonable?

Statistics: Probably using a GLM with a repeated measurement design with all the six plots together provides more robust statistical outcome than separating the individual slope positions. The pre-treatment measurements are valuable and add

some robustness, but they were restricted to 9 days.

Together with the root data in the supplements, this is a really highly interesting story, but due the above mentioned limitations, probably not strong enough for Nature Communications.

Reviewer #2

(Remarks to the Author)

This manuscript reports the results of a one-year in situ warming experiment in a tropical forest. This kind of experiment is logistically challenging to do, by doing this in a tropical forest this study makes a novel and valuable contribution. The study finds that the response of soil respiration to increased temperature is higher than reported in temperate forest studies, and that this response is especially strong in one of the three pairs of plots. Despite this response, temperature sensitivity is lower in the heated plots, indicating that either the soil respiration response has acclimated or that there are limits to adaptation and respiration cannot continue to increase with warming temperatures.

I do feel the framing and presentation of the study could be more nuanced about what this study adds to the literature. There is an implicit assumption that in situ experiments are the best way to find out how tropical forests will respond to warming, and that other approaches (observational studies such as space for time substitutions, lab incubations) are second best. I disagree with this, and think we can learn interesting things by comparing what different methods tell. For example, a space-for-time substitution such as an elevation transect captures responses after the system has had time to adapt (e.g. giving time for shifts in community composition), so may capture mechanisms that are evident under long-term press changes but not pulse changes. By comparing what different methods tell us we can get a more complete understanding of what is going on. See Smith et al. 2009 Ecology (https://www.jstor.org/stable/25660976?seq=1#metadata_info_tab_contents) for a theoretical perspective on how responses to environmental change are expected to vary with time, which is useful for placing this study in context.

Associated with the point above, while the manuscript is generally well written, the language is sometimes a bit overblown, e.g. line 97 "the extreme response of the Upper Slope site was unprecedented". To me it seems to be trying too hard to emphasise the unexpected nature of the results.

In some places aboveground and belowground responses of tropical forests are confused. For example, in lines 73-75 the strong temperature sensitivity of the results of this study are contrasted with the weaker temperature responses found by a recent study of spatial variation in carbon (ref 12, Sullivan et al.), yet that study focused exclusively on aboveground carbon stocks and not on soils.

I have one technical concern with the manuscript. Some of the statistical analyses appear to include replicate data from the same plot taken at different times without accounting for the likely non-independence of the observations (e.g. the sample size quoted for the t-tests seem very high given for what I understand the size of the experiment is, so there must be some pseudoreplication). This should be addressed by using statistical analyses that account for the non-independence of observations (e.g. repeated measures ANOVA, mixed effects models, GLS with correlation structure accounting for temporal autocorrelation), or averaging repeated measurements so that all the data that go into the analysis are independent replicates. Looking at the graphs, I feel convinced that the differences revealed by the pseudoreplicated analyses are real, but the analysis should attempt to deal with pseudoreplication, or at least put more emphasis on estimating the effect sizes rather than statistical significance.

There is quite a lot of discussion justifying the inclusion of 'hot moments' (lines 77-87), i.e. occasional very high respiration values. I am convinced by the argument for including them, but it would be interesting to know the extent they drive the differences between treatments and the high warming response (e.g. add an extended data figure excluding these).

A few detailed comments:

Line 104: Why do you think fine root mass increased in control plots? Was there disturbance when the plots were established?

Line 110: The argument for why you think it is not roots could be clearer. Is it that the difference in roots is in the wrong direction for it to be driving the respiration differences?

Lines 272-274: Wouldn't it be simpler to analyse this with a paired t-test?

Line 274: One-sample t-test, not one-way.

Figures 1 and 2: As there are only three plots, it would be great to show the data from each rather than grouping the lower and middle plots. This way all the data can be displayed, and the figures would only have to be a little more complex.

Figure 3: Interesting that the upper plot seems to have a slightly higher Q10. I wonder if there is some interaction between being pre-adapted to warmer conditions (i.e. being lower down) and reduced sensitivity to temperature. With three pairs you can't test this, but you could help a future meta-analysis by providing the typical soil temperatures along with the Q10s.

Version 1:

Reviewer comments:

Reviewer #1

(Remarks to the Author)

How to start. The revised manuscript profits a lot from the improved statistics and authors re-visited the site to check the spatial variation of soil respiration at the remote field site. This for sure is more than highly appreciated. And indeed, the spatial variation was not very high (at least during the few days of additional measurements) indicating that spatial variability likely was not the main cause of the sporadically extremely high CO₂ fluxes which were detected at a certain warmed plot. It is a pity that a hurricane destroyed the infrastructure and somehow hampered the planned experiments. Otherwise, I am sure that the measurements would have been performed for a longer period and more auxiliary measurements would have been made. Overall, I am still sceptical that the outcome of this study is robust enough to be published in such a high impact journal.

Pros:

The setup of the soil warming in a tropical forest is unique. This is one out of two artificial warming experiments in the south American Tropics – hence the outcome is in any case relevant for the scientific community, modellers, etc. Overall, the CO₂ flux was much higher in warmed plots, which is a clear result.

Cons:

The study basically shows soil CO₂ efflux data from six chambers (3 warmed, 3 control). One of the three warmed-plot chambers periodically showed extremely high soil CO₂ efflux. These extremely high CO₂ effluxes could not be explained with the measured environmental and soil or root variables and hence, a reasonable mechanistical explanation is lacking. CO₂ fluxes were measured in very high temporal resolution (half hourly), but there is no deeper look into this high-resolution data. It would be interesting if the extremely high fluxes occurred randomly or if they were part of diurnal patterns or related to events such as rainfall. Authors thoroughly checked the instrumentation and tried everything to avoid measurement bias. This is now provided in the supplements (I only wonder that they did not observe high CO₂ fluxes when they “breathed directly into the instrument” – when I breath into my IRGA, the CO₂ concentration increases immediately to >10.000 ppm...). However, the most important information about chamber closure times and air flow, tubing length etc. during field measurements are nowhere to find in the ms.

Soil moisture was an important explanatory variable of the temporal variability of the soil CO₂ efflux. The plot with extremely high CO₂ fluxes was permanently wetter than the corresponding control plot. For me there is no logical explanation that artificial understory and soil warming causes an increase in soil moisture. Typically, moisture decreases under warming or remains similar if repeated rain rewets the soil. Therefore, apparently, the much higher soil moisture in this particular plot was independent of the warming treatment – but might have had implications on the CO₂ efflux as shown in Suppl. Fig. 4. While control and warming plots at the two other treatment sites showed a similar response to moisture and temperature together, the control and warming plot at the upslope site (the one with extremely high emissions) showed a very different pattern. Control plot (5) soil CO₂ efflux was basically driven by soil temperature, while warmed plot (6) CO₂ efflux was driven by both temperature and moisture. This also suggests that soil moisture, which can be considered as independent of the soil warming treatment, had a huge effect on the high CO₂ fluxes from this plot. With a replication of only n=3 plot pairs, a single plot, which could have been influenced by other parameters as warming is quite critical.

In the concluding paragraph L285-287 it is stated that: “Nevertheless, understanding the underlying mechanisms driving the response of soil carbon loss to warmer temperatures is critical...” - the current study does not provide these mechanisms (and a soil carbon loss was not detected either).

There is gap of two whole months (Feb, Apr) in Fig 1 and 3 showing the soil CO₂ efflux. I could not find any explanation for this data-gap in the text. In L319 it is stated that equipment run almost continuously except during November.

The paired plots showed somewhat contrasting temporal response patterns to warming. Lower slope responded positively during the first months but responded already negatively to warming during the last months. Mid slope plot showed a negative response during some of the early phase but strong positive response later. Upslope plot also needed some time until it really increased dramatically. These contrasting temporal patterns could be discussed.

The conclusion in Line 282 that “from warmed plots with no signs of declining in the first 1-2 years after initiating warming” is and overstated. Soil was warmed only during a single year. How to conclude on the second year?

Reviewer #2

(Remarks to the Author)

In the original submission, reviewer 1 and myself raised concerns that (1) low replication meant that random spatial variation could explain apparent treatment differences, (2) analyses were pseudoreplicated and (3) the high soil respiration values were hard to explain and potentially implausible. The authors have gone to considerable effort to address these, with new field data and extensive new analysis. In relation to the original concerns:

1. The new field data on spatial variability helps to put this concern in context. While of course having more spatial replicates and greater spatial intermingling of treatments would be ideal, this new data does help (a) make the reader explicitly aware

of the nature of the risk and (2) provide some reassurance that it doesn't drive patterns.

2. The new GLS analysis is helpful and reassures me that differences aren't driven by treating all observations as independent.

3. I appreciate the extensive checks to make sure that the high readings were unlikely to be artefactual. Ultimately I don't think we always need to be able to explain unexpected data as long as we are confident the data are real.

In sum I think the original issues have been addressed or at least raised as transparent limitations.

I have two minor suggestions to consider:

- I am struck by the increased variability at higher temperatures, but discussion largely focuses on the mean. Perhaps a sentence or two on the implications of increased variability would be helpful.
- To appreciate this variability, I would find it helpful in Fig. 2 to put all the graphs on the same y axis scale.

Warming induces unexpectedly high soil respiration in a wet tropical forest

Tana E. Wood^{1*}; Colin Tucker^{2,3}; Aura M. Alonso-Rodríguez^{4,5}; M. Isabel Loza⁶; Megan Berberich⁷; Iana Grullon Penkova¹; Molly A. Cavaleri⁷; Christine S. O'Connell⁸; Sasha C. Reed²

Response to Reviewers

REVIEWER COMMENTS

Reviewer #1 (Remarks to the Author):

Tana Wood and co-authors present data from a pioneering soil warming study in the tropics. To my knowledge this is the first manipulative warming study in the tropics and the results are of high interest since modelling exercises suggested that warming effects on soil and ecosystem carbon cycling could be especially responsive in these (warm-wet) ecosystems. Authors basically present a one year time series from six auto-chambers measuring soil CO₂ efflux and observed a strong positive response of soil CO₂ efflux to artificial warming. Though I think this study gives some strong indications, I am not convinced that the broad conclusions are justified by the dataset. The simple reason is the inefficient spatial replication of soil CO₂ efflux measurements.

We thank the reviewer for recognizing the value of the study. The warming experiment presented here is the culmination of years of development so that we would have the opportunity to address critical questions on the response of tropical forested ecosystems to warmer atmospheric temperatures.

We fully acknowledge that soil respiration is spatially heterogeneous and that the trade off between our dataset's high temporal resolution (half-hourly) and relatively limited spatial resolution (one autochamber replicate per plot) is a challenge. We thank the reviewer for this high-level comment as we believe that our efforts to track down the level of spatial representation in our soil respiration dataset led to dramatic improvements in the manuscript.

We have addressed this concern using several statistical analyses and by conducting an additional field campaign of data. From our updated methods:

“A further 152 flux measurements were taken in fall 2020 (from November 4-19, 2020) across the full TRACE field site in order to assess spatial heterogeneity in soil CO₂ flux rates and characterize whether the rates seen in the six experimental TRACE plots were representative of the overall spatial heterogeneity. We collected CO₂ measurements from 30 collars installed outside the TRACE plots, and from the 6 collars installed previously in each experimental plot. We took 4 measurements from most collars located outside of the plots, 5 measurements from the collars inside of the plots, and 3 observations from 2 of the collars outside of the plots due to inclement weather during field work. Spatial interpolation was used to create a flux “spatial variability” map (Supplemental Figure S2).”

We describe additional field and equipment checks to validate our high fluxes in Section A of the Supplemental Methods.

We interpreted our spatial survey results as such (lines 124-139, main text, also including the new statistical checks):

“We used several approaches to confirm that our observed emissions increases were indeed primarily driven by warming and not by stochastic spatial variability or measurement error (Supplemental Materials and Methods). Whereas the Upper slope is clearly a “hotspot” for higher soil respiration rates, throughout the study, all of the plots exhibited extreme “hot moments” of soil respiration, where observed flux values were several times higher than the mean value (Figure 3). We further conducted a spatial survey of 30 locations randomly selected outside of the plot locations across the 40 m x 60 m TRACE area to quantify the spatial variability of soil CO₂ fluxes across the TRACE landscape (Supplemental Figure S2). We unsurprisingly found variability across space but no evidence that the warm plots are systematically located on landscape-level hot spots. Additionally, pre-treatment data showed no statistical differences in soil respiration rates between paired warmed and unwarmed plots prior to the initiation of warming (Supplemental Figure S3), and running a generalized least squares model with and then without the Upper slope (i.e., the control-warming paired plots with particularly large differences in CO₂ flux rates) did not change the finding that warmed plots had significantly higher soil respiration rates (Supplemental Table S1, Supplemental Materials and Methods). Field sensitivity analyses failed to detect evidence that invertebrate or animal presence in the chamber could have produced such high respiration rates (Supplemental Materials and Methods).”

Finally, we drew some of our conclusions from previous soil respiration and trace gas flux work in this system that, taken together across studies and sites within this forest, support our assertion that in our system high temporal resolution soil respiration data that considers topographic position is more relevant for detecting treatment responses than including high spatial resolution within each topographic location. Further, these high temporal resolution data are critical for examining the dynamic responses of soil respiration to environmental variables that are also highly variable temporally, such as soil moisture and temperature. The prior work informing our interpretations of these results include:

- In Wood and Silver 2012, CO₂ was sampled bi-weekly from five static chambers that were droughted and five that served as controls in each of three topographic positions (Ridge, Slope, Valley; 10 plots per topographic position). This study found significant differences in soil respiration rates among topographic positions, but small variation in rates within.
- However, bi-weekly measurements did not detect treatment effects of drought in the valley location while a drought response was detected in the ridge and slope locations. However, in addition to bi-weekly static chamber measurements, soil respiration was also measured hourly with 6 automated chambers in the valley location with three that received drought and three that were controls (Wood et al. 2013). With these high temporal resolution data we were able to detect a drought response in the valley location and in addition, we were able to describe relationships between soil respiration and moisture as well as temperature.

- In a later study in the same forest, O'Connell et al. 2018 measured soil respiration with three automated chambers in each of three topographic positions (Ridge, Slope, Valley). They found large variation in soil respiration rates among topographic positions, but small variation within with the highest fluxes found on slopes (e.g., slope = 3.794 +/- 0.688 umol/m²/s; valley = 2.833 +/- 0.065 umol/m²/s). They additionally found hot moments of soil respiration to be important.

Taken together, these prior studies support our assertion that, in our system, when under resource and equipment constraints, high temporal resolution soil respiration data that considers topographic position is more relevant for detecting treatment responses than prioritizing high spatial resolution within each topographic location (though of course both would be ideal!). Further, these high temporal resolution data are critical for examining the dynamic responses of soil respiration to environmental variables that are also highly variable temporally, such as soil moisture and temperature.

Specific comments:

Soil CO₂ efflux typically shows high spatial variation. In the presented study, soil CO₂ efflux was measured at totally 6 spots (3 chambers at control plots, 3 chambers at warmed plots; n =3). On top, cambers (plots) were aggregated along slope positions (lower slope, middle slope, upper slope; n=1) with regard to statistics and the interpretation of the results. Accordingly this is the lowest amount of spatial replications possible for doing any statistics but certainly does not cover much spatial heterogeneity in fluxes. Authors suggest that the individual auto-chambers produced high temporal resolution measurements (several thousand flux estimates per year) and that the paired-in-time-differences between the treatment and control chambers can be used as kind of time in space substitution. This is not so convincing to me. Even with some thousand temporal replicates of a single chamber one cannot exclude that (I) any chamber showed some temporal malfunctions and (II) that unexpected other sources temporarily contributed to the CO₂ production in single chambers. The extremely high CO₂ efflux at the warmed upslope chamber might be such an example. The high emissions are difficult to explain except that there was some chamber malfunction (indicator would be that there were only half as much pairs upslope as mid- and downslopes, L59-62, Fig2) or that some insects, mammals, or whatever, temporarily occupied the chamber and contributed to the CO₂ production (indicator for that would be the rather erratic spikes in CO₂ differences of up to 400,mmol in fig 2 for the upper slope position – such spikes can hardly be explained with enzyme kinetics of decomposer microbes).

The reviewer raises a valid question. We too were concerned that the upper location and the hot moments that we observed were the result of alternate sources of CO₂ (such as animals) or that they could be due to equipment malfunction. As such, we invested considerable effort into validating our data. We describe the series of tests that we undertook below:

1. At a moment when one of the plots was exhibiting higher than expected fluxes, we tested the full factorial of equipment configurations to compare a plot that was demonstrating high fluxes with a plot that was demonstrating low fluxes:
 - a. Switched the chamber that exhibited high fluxes with a chamber that exhibited low fluxes
 - b. Changed the cable of the site with high fluxes and that of low fluxes

- c. Used a different port under the above configurations to determine if the port was the issue.
- d. Changed out all filters and checked all tubing.

Regardless of the configuration we used (i.e., new chamber, new cable, different port) the fluxes remained high in the high flux site and low in the low flux site.

2. In addition to testing the configuration, we also flushed all cables with 70% isopropyl alcohol per LiCor recommendation, to flush out any mold or insects that might have entered the cables that could be contributing to higher fluxes. Even after flushing cables, we continued to observe high fluxes.
3. One of the “malfunctions” that can occur with automated chambers is a failure to seal or close completely. We can identify these values by looking at the coefficient of variation of the individual measurements. As such, when we have a CV that is greater than 2% we look at the raw data and can evaluate whether the CO₂ measurements increase linearly as we would expect. When this is not the case, we excluded the data.
4. As the reviewer mentioned, animals could also contribute to the high fluxes observed. We visually inspected all installed chambers and collars during the period of high fluxes and did not observe any frogs, lizards, ants, or other organisms that could contribute to the high fluxes observed. We additionally conducted tests where we manually blew CO₂ into the LiCor IRGA and even with this test, we could not accomplish the high fluxes that we observed in the field. Nevertheless, we inspect the collars for branches that could impede closure and organisms as part of our weekly maintenance. We also trim seedlings that could also interfere with CO₂ measurements. Given that we have not observed animals in the collars, and the fact that a large adult male breathing directly into the instrument did not achieve the values observed, we concluded that it is highly unlikely the fluxes were driven by the small organisms that are found in our study area (frogs, lizards, snails – mongoose are the only mammal occasionally observed in our site), even temporarily.
5. As an additional confirmation that our data were valid. We evaluated the background CO₂ to determine if these values were within the expected range. We additionally examined raw data to see that from the moment measurements were made that the CO₂ increased linearly within the chamber as would be expected. Further, we sent our raw data to LiCor and had their scientists examine our data and they also concurred that after considering all of the possible reasons described above that the high fluxes we were observing were indeed real.

Well, such a kind of outlayer-chamber is not really a problem if it is only one of many spatial replicates. It, however, becomes a real issue if there is only one chamber available, such as it was the case here.

We understand the reviewer's concern that the results are driven by a single outlier that could be characteristic of this particular location and not a direct response to warming. We have reanalyzed the data via a GLS model accounting for temporal autocorrelation within a given chamber, and the results

are qualitatively unchanged. Furthermore, exclusion of the chamber with extreme values within this GLS analysis did not change the overall treatment effect (although it did of course eliminate the treatment x hillslope interaction term significance). We now only present the paired difference as an estimate of the magnitude of the warming effect, and focus our statistical analysis on the GLS and Bayesian approaches.

With regard to the assertion that the spikes cannot be driven by enzyme kinetics of decomposer microbes, we completely agree. This paper is not a presentation of heterotrophic respiration dynamics, but of soil respiration (i.e., soil surface effluxes) that reflect a suite of potential ecosystem-scale drivers, such as soil moisture, roots, soil texture, and we do not claim that enzyme kinetics alone explains the observed response.

There is some general contradiction in the argumentation. It is suggested that the temperature sensitivity of the soil CO₂ efflux decreases with the 4°C warming (from about 2 to about 1). How can then a 4°C warming lead to a 200% increase in soil CO₂ efflux? The Q₁₀ would be far beyond.

To address this point, we conducted an analysis analysis of different approaches to estimating Q₁₀, to calculate the following three methods:

- 1) evaluating Q₁₀ within a given chamber as a response to the soil temperature within that chamber both via a GLS model and a hierarchical Bayesian, within convergent although quantitatively different estimates of Q₁₀ between the control and warmed plot
- 2) evaluating the effective Q₁₀ based on the simultaneous changes in temperature and soil moisture between a pair of control and warmed plots using a model based calculation
- 3) evaluating the cumulative Q₁₀ by comparing the cumulative respired C and the average temperature difference between control and warmed plots.

Our qualitative story that temperature sensitivity was lower in the warm plots was robust to these distinct analyses.

The reviewer is right to point out that the instantaneous Q₁₀ (the most rigorous version of the temperature sensitivity metric) implies a different response than the cumulative difference between chambers. This is not a simple miscalculation, the within-chamber estimates of temperature sensitivity are robust to the analytical method we applied, and the cumulative effects show a much greater response to the warming treatment than the instantaneous temperature sensitivity, which suggests that it is not simple enzyme kinetics, but a regime shift in ecosystem processes driving the large response. Furthermore, while the magnitude of this difference is very large in this study, it is consistent with previous observations of declining temperature sensitivity with warming. We point out here that the 3-4C warming we imposed in this study is within the bounds of many other warming experiments, but it is a much greater warming relative to the annual temperature range at the study site, when compared with any other terrestrial warming experiment we are aware of. Thus, we argue that this warming effectively pushed the ecosystem into an almost complete novel set of soil microclimate conditions.

It is argued that the high temporal resolution allowed covering “hot moments” in soil respiration. Indeed there were fluxes up to 400 μ mol but can such fluxes be explained with any logical enzyme kinetics or whatever related to SOM decomposition response the 4°C increase in soil temperature? Q₁₀ extrapolated from seasonal data always needs to be taken with caution as many other factors aside soil temperature affect the temporal course of soil CO₂ efflux.

We agree that it is not likely enzyme kinetics or decomposition responses alone that are driving the observed patterns. While that may indeed be part of it, we are observing soil surface CO₂ effluxes, not heterotrophic respiration per se. That said, the high rates we observed are really challenging to explain. We note that since this paper entered its initial review, a second tropical warming manipulation in the tropics also reported exceedingly high soil respiration rates with warming (Nottingham et al. 2020) and also struggled to explain the entirety of that increase (Nottingham et al. 2022). Together, TRACE and SWELTR are building a body of experimental results that open up numerous additional questions about how the tropical terrestrial carbon cycle may respond to warming.

The reviewer is correct to assert that Q₁₀ from seasonal data may incorporate phenological signals, etc, in a way that may be confusing. We point out that in our original hierarchical Bayesian approach, Q₁₀ was estimated within each month for a given chamber, and then an overall Q₁₀ for that chamber was estimated from those monthly Q₁₀s, and the treatment effect on Q₁₀ was then estimated from the chamber Q₁₀. This approach mitigated the possibility of mistaking a phenological signal for a temperature sensitivity signal. Subsequently, we also evaluated the data using a generalized least squares approach, which allowed us to estimate Q₁₀ across all the sampling dates (but not hierarchically as in the Bayesian approach) and we came to very similar estimates of the Q₁₀ difference between treatments. We focus our analysis of the Q₁₀ on the Bayesian results because we agree that mitigating the potential seasonal confounding effect is the best practice.

Finally, we went to every effort to evaluate those data and ensure that they were not simply errors in curve fitting, contamination of the sampling lines, animal respiration within the chambers, or another explanation.

If warming really induced a loss of 96 Mg C ha⁻¹ y⁻¹ at the upslope site, then the whole soil C (numbers are not reported in the paper) of the site might be respired within a few years. Is this reasonable?

This is an excellent question. Conceivably, if we consider this experiment in the context of results from high latitude ecosystems, then we would eventually expect soil respiration values to “acclimate” with time and return to control values. Given rapid rates in tropical ecosystems, we would expect this acclimation to occur earlier than has been observed in those systems. We are currently working with global modelers to pull together multiple data sets that we have been collecting as part of this experiment to explore future scenarios and assess how long these rates can be sustained. This is in addition to continuing to collect long-term measurements.

Statistics: Probably using a GLM with a repeated measurement design with all the six plots together provides more robust statistical outcome than separating the individual slope positions. The pre-treatment measurements are valuable and add some robustness, but they were restricted to 9 days.

We agree, as did the other reviewer, and we have taken the reviewers' suggestions into account and reanalyzed the data using a generalized least squares approach accounting for temporal autocorrelation of fluxes within a given chamber, as well as better dealing with heterogeneity. The results support our previous assertion, and add some nuance as well. Thank you for this suggestion.

Together with the root data in the supplements, this is a really highly interesting story, but due the above mentioned limitations, probably not strong enough for Nature Communications.

We understand the concerns raised by the reviewer and hope that they agree that, with the additional data included, revised statistical analyses, and detailed explanations of the methodological tests that we ran, the manuscript is now significantly stronger and will contribute to a quickly changing field of broad interest to the *Nature Communications* readership.

Reviewer #2 (Remarks to the Author):

This manuscript reports the results of a one-year in situ warming experiment in a tropical forest. This kind of experiment is logistically challenging to do, by doing this in a tropical forest this study makes a novel and valuable contribution. The study finds that the response of soil respiration to increased temperature is higher than reported in temperate forest studies, and that this response is especially strong in one of the three pairs of plots. Despite this response, temperature sensitivity is lower in the heated plots, indicating that either the soil respiration response has acclimated or that there are limits to adaptation and respiration cannot continue to increase with warming temperatures.

We thank the reviewer for their summary of this study and for their feedback below. We believe that the changes we have made to the manuscript in response have substantially improved the paper, and appreciate comments from Reviewer 1 and 2.

I do feel the framing and presentation of the study could be more nuanced about what this study adds to the literature. There is an implicit assumption that in situ experiments are the best way to find out how tropical forests will response to warming, and that other approaches (observational studies such as space for time substitutions, lab incubations) are second best. I disagree with this, and think we can learn interesting things by comparing what different methods tell. For example, a space-for-time substitution such as an elevation transect captures responses after the system has had time to adapt (e.g. giving time for shifts in community composition), so may capture mechanisms that are evident under long-term press changes but not pulse changes. By comparing what different methods tell us we can get a more complete understanding of what is going on. See Smith et al. 2009 Ecology (https://www.jstor.org/stable/25660976?seq=1#metadata_info_tab_contents) for a theoretical

perspective on how responses to environmental change are expected to vary with time, which is useful for placing this study in context.

We wholeheartedly agree with the reviewer that observational studies are incredibly valuable, and the authors of this manuscript often utilize natural gradients and long-term data to address our research questions. Experiments are simply a tool that allows us to generate controlled conditions that isolate specific variables as well as to generate conditions that do not currently exist for a particular ecosystem. What we accomplish with this design that cannot be recreated with gradients is to generate warmer conditions in ecosystems that are already consistently warm, thus pushing them into a novel climate envelope, which can augment what we can learn from lab studies, elevational gradients, and other approaches.

We have revised the text (lines 50-69) to provide more nuance surrounding how these different approaches can provide different types of information about ecosystem responses to atmospheric warming.

Associated with the point above, while the manuscript is generally well written, the language is sometimes a bit overblown, e.g. line 97 “the extreme response of the Upper Slope site was unprecedented”. To me it seems to be trying too hard to emphasise the unexpected nature of the results.

We have edited language throughout (including this example from Line 97 of the initial submission) to lower the temperature on this kind of language; thank you.

In some places aboveground and belowground responses of tropical forests are confused. For example, in lines 73-75 the strong temperature sensitivity of the results of this study are contrasted with the weaker temperature responses found by a recent study of spatial variation in carbon (ref 12, Sullivan et al.), yet that study focused exclusively on aboveground carbon stocks and not on soils.

We appreciate the reviewer drawing our attention to this and have revised this section to ensure clarity. We do agree that the study of Sullivan et al. 2020 focused on aboveground responses of carbon; however, the authors include the following statement, which highlights the broader thinking that tropical soils are not likely to exhibit a strong temperature response, in this case, due to concomitant changes in soil moisture availability:

“Notably, the temperature-precipitation interaction we found for aboveground stocks is in the opposite direction to temperature-precipitation interactions reported for soil carbon (25). In soils, moisture limitation suppresses the temperature response of heterotrophic respiration, whereas in trees, moisture limitation increases the mortality risks of high temperatures.”

I have one technical concern with the manuscript. Some of the statistical analyses appear to include replicate data from the same plot taken at different times without accounting for the likely non-

independence of the observations (e.g. the sample size quoted for the t-tests seem very high given for what I understand the size of the experiment is, so there must be some pseudoreplication). This should be addressed by using statistical analyses that account for the non-independence of observations (e.g. repeated measures ANOVA, mixed effects models, GLS with correlation structure accounting for temporal autocorrelation), or averaging repeated measurements so that all the data that go into the analysis are independent replicates. Looking at the graphs, I feel convinced that the differences revealed by the pseudoreplicated analyses are real, but the analysis should attempt to deal with pseudoreplication, or at least put more emphasis on estimating the effect sizes rather than statistical significance.

We have taken the reviewer's suggestion into account and reanalyzed the data using a generalized least squares approach accounting for temporal autocorrelation of fluxes within a given chamber, as well as better dealing with heterogeneity. The results support our previous assertion, and add some nuance as well. This was very helpful feedback and led to an increased level of confidence in the observed trends.

There is quite a lot of discussion justifying the inclusion of 'hot moments' (lines 77-87), i.e. occasional very high respiration values. I am convinced by the argument for including them, but it would be interesting to know the extent they drive the differences between treatments and the high warming response (e.g. add an extended data figure excluding these).

We have now included Supplemental Table S1. Generalized least squares model regression coefficients excluding upper slope chambers with extreme values. This Supplemental Table reports the model run in which we completely excluded the upper hillslope with exceptionally high values from the GLS analysis. In this model run, the warming effect remained, although unsurprisingly the warming x location effect was no longer significant.

We think this is a very helpful addition to the Supplement and thank the reviewer for this idea.

A few detailed comments:

Line 104: Why do you think fine root mass increased in control plots? Was there disturbance when the plots were established?

We have now further evaluated root dynamics in our plots with two publications since submission of this manuscript (Yaffar et al. 2020; Tunison et al. 2023). We took great care to minimize disturbance during installation and both warmed and control plots received the exact same level of disturbance during installation of the plots as well as minirhizotron tubes and ingrowth cores. There are seasonal dynamics to root growth, but the effect of warming far outweighed the temporal patterns.

Line 110: The argument for why you think it is not roots could be clearer. Is it that the difference in roots is in the wrong direction for it to be driving the respiration differences?

We have since published a manuscript that focuses on the response of root specific respiration to warmer temperatures (Tunison et al. 2023). We found no significant acclimation of root specific

respiration to warmer temperatures; however, when we combine these results with estimated root biomass in the plots, we find a reduction in total contribution of roots to total soil respiration in the warmed plots relative to the controls. At the same time, we find that when warming is active, we have a 50% increase in microbial biomass carbon. This suggests that the increase in soil respiration to warmer temperatures is likely primarily microbially driven.

We have included additional text in the manuscript to further explain this.

Lines 272-274: Wouldn't it be simpler to analyse this with a paired t-test?

Our thinking was that constructing the pairs in time and then testing whether differences are > 0 would allow us to "match" conditions across both space and time in the data series. We have left this analysis as is, but are open to altering it at this point if asked during an additional round of reviews.

Line 274: One-sample t-test, not one-way.

Fixed; thank you (Line 355).

Figures 1 and 2: As there are only three plots, it would be great to show the data from each rather than grouping the lower and middle plots. This way all the data can be displayed, and the figures would only have to be a little more complex.

The figures have been remade to separate all three plots.

Figure 3: Interesting that the upper plot seems to have a slightly higher Q10. I wonder if there is some interaction between being pre-adapted to warmer conditions (i.e. being lower down) and reduced sensitivity to temperature. With three pairs you can't test this, but you could help a future meta-analysis by providing the typical soil temperatures along with the Q10s.

Thank you for this thought and we are in strong agreement about open data principles. Upon publication, all included data will become freely available via the Dept. of Energy ESS data portal. With respect to adding a specific table within the supplement, this site doesn't vary meaningfully in temperature across topography, as the overall altitudinal change is on the order of only tens of meters. Thus, the mean annual temperature reported is applicable to all three topographic zones. We hope that we clarified this in the revised version of the manuscript and methods and that our data availability addresses this concern.

Response to Reviewers

Thank you again for submitting your manuscript "Warming induces unexpectedly high soil respiration in a wet tropical forest" to Nature Communications. We have now received reports from 2 reviewers and, based on their comments, we have decided to invite a revision of your work. Your revision should address all the points raised by our reviewers (see their reports below). In particular, for your work to be properly assessed, mechanisms driving observations should be presented and additional methodological details are required.

When resubmitting, you must provide a point-by-point response to the reviewers' comments. Please show all changes in the manuscript text file with track changes or colour highlighting. If you are unable to address specific reviewer requests or find any points invalid, please explain why in the point-by-point response.

Thank you for the opportunity to revise and resubmit the manuscript. We have addressed each point raised by the reviewers in bold font below each of the points raised. Included in our response, we provide additional information on mechanisms that could be driving observed results and include the additional methodological details requested.

Important: In addition to the above, you must comply with the following editorial requests; we will not be able to proceed with your revised manuscript otherwise. Please also see the *Nature Communications* formatting instructions, which you may find useful while preparing your revised manuscript.

We have adhered to all Nature Communications formatting instructions.

REVIEWER COMMENTS

Reviewer #1 (Remarks to the Author):

How to start. The revised manuscript profits a lot from the improved statistics and authors re-visited the site to check the spatial variation of soil respiration at the remote field site. This for sure is more than highly appreciated. And indeed, the spatial variation was not very high (at least during the few days of additional measurements) indicating that spatial variability likely was not the main cause of the sporadically extremely high CO₂ fluxes which were detected at a certain warmed plot. It is a pity that a hurricane destroyed the infrastructure and somehow hampered the planned experiments. Otherwise, I am sure that the measurements would have been performed for a longer period and more auxiliary measurements would have been made. Overall, I am still sceptical that the outcome of this study is robust enough to be published in such a high impact journal.

Thank you for your assessment. We agree that our providing of additional data on spatial variability improved the manuscript substantially. We genuinely appreciated this suggestion from reviewers and believe these new data help rule out the alternative hypothesis that the results were driven by spatial variability rather than the warming treatment.

As the reviewer acknowledges, Hurricanes Irma and Maria in 2017 were devastating for the warming infrastructure, which took a full year to repair. We agree that having that additional time to collect data would have been ideal, but as we are sure the reviewer understands, climate disasters are an inherent risk involved in conducting field research. In addition to creating challenges scientifically, the people conducting the research were also set back by these events. The local team, including the lead author, were without basic resources such as power and water, for months following the storms. However, we have persevered and were able to rebuild the experiment, and we have continued to collect data to better understand the results presented in this manuscript. We hope that the revisions and our response to reviewers will help resolve the remaining reservations of reviewer 1, and we highlight that this remains the only tropical forest warming experiment of its kind. We feel the unique nature of these data and the insights they provide will be of interest to a broad readership.

We also appreciate the reviewer's preference for high replication and for setting a high bar. When designing a field experiment, trade-offs are inevitable. For this experiment we determined that it was critical to incorporate warming of both vegetation and soils and to capture high temporal variability, which unavoidably limited our ability to include more than three replicates for multiple reasons. While the project was interrupted by the hurricanes, our approach allowed us to rebuild and to support longer-term data collection. In the almost 10-years since we broke ground on the warming experiment only one additional tropical field warming experiment has come online, which is the soil warming experiment in Panama that we discuss in the manuscript. Like our experiment, the Panama warming experiment has also had substantial gaps in their application of the warming treatment, even with their use of lower-maintenance infrastructure and with much less frequent (every two weeks instead of half-hourly, an enormous difference in temporal resolution) soil respiration measurements that they collected manually. Accordingly, we are the only experiment in the world that has the capacity to provide high temporal resolution soil respiration responses to warmer temperatures in a wet tropical forest, and that includes data such as, for example, nighttime respiration rates. Wet tropical forests comprise the largest fluxes of soil respiration to the atmosphere of any terrestrial ecosystem and as a result drive a significant proportion of the uncertainty in our ability to predict feedbacks to future warming. That our results are surprising with warming effects greater than expected underscores the value of data from these ecosystems, and the unique position that this project is in to provide them. We include detailed responses below that we hope will alleviate any remaining concerns.

Pros:

The setup of the soil warming in a tropical forest is unique. This is one out of two artificial warming experiments in the south American Tropics – hence the outcome is in any case relevant for the

scientific community, modellers, etc. Overall, the CO₂ flux was much higher in warmed plots, which is a clear result.

We appreciate that these new data helped convince the author that the results are indeed due to the warming treatment. As the reviewer mentions, we are one of just two such experiments in any tropical ecosystem and we are the only experiment that is warming both vegetation and soils, which allows the incorporation of above and belowground processes, making this project unique. Further, we are currently working with modelers that need high temporal resolution data to better develop predictive models for the response of soil respiration to warming in the tropics. While the experiment conducted in Panama had more replication of their warming treatment (5 plots versus 3), they only collected soil respiration data every two weeks using static chamber method, which means they were not able to capture the highly dynamic nature of soil respiration through time. This is not to diminish the value of the work conducted by this group, but to illustrate the added value and capacity of the TRACE project to significantly advance our understanding of these understudied ecosystems.

Cons:

The study basically shows soil CO₂ efflux data from six chambers (3 warmed, 3 control). One of the three warmed-plot chambers periodically showed extremely high soil CO₂ efflux. These extremely high CO₂ effluxes could not be explained with the measured environmental and soil or root variables and hence, a reasonable mechanistical explanation is lacking.

As the reviewer points out, one warmed plot demonstrated extremely high fluxes and, per this reviewer's suggestion, we now more clearly describe reasonable mechanistic explanations (more on that in the subsequent paragraph) However, we would like to highlight that all of the warmed plots showed higher than expected soil CO₂ efflux rates, even when the high values from the upper slope plots were excluded. What this shows is that even with the variation in the magnitude of the response, the positive effect of increased temperature on soil respiration is consistent and highly robust.

We agree that including hypotheses for mechanisms that could explain the especially high fluxes found in our upper slope plots relative to the mid slope and lower slope plots would improve the manuscript and have now included additional text to provide potential mechanisms that could be driving this response. Please see lines 127-131 where we discuss the prevalence of "hot moments" in our system. Please also see lines 148-154 where we discuss possible mechanisms that could drive the higher fluxes on the upper slope.

As the reviewer mentions, soil moisture was higher in the warmed relative to the control in the upper slope plots and that these were also the plots with the strongest response to warming. We agree that the expectation is for there to be drying of soils in response to warmer temperatures, but our consistent increases in soil moisture with warming offer exciting insight into the unpredictable ways these systems may respond to global change. One hypothesis for how this can occur is that the forest floor litter layer dries substantially under warmer conditions, thus slowing decomposition and creating a buffer (due to higher litter mass) against evaporative water loss from the soil. An experiment at a nearby site in Puerto Rico that trimmed the forest canopy and added the debris to the forest floor found that,

despite warmer temperatures (due to less canopy shading), soil moisture *increased* in these sites due to the buffering effect of this additional litter layer. This supports the idea that diverging responses of different ecosystem compartments could directly and indirectly affect soil moisture and associated process rates.

In considering how these plots differed, sloped plots are more likely to have a thinner litter layer than on an upper slope or ridge due to movement and runoff. Fungal mats have been found to be critical for connecting the litter layer to the forest floor, thus minimizing erosion. In our site, fungi are negatively affected by warmer temperatures and drier conditions (Puentes in review), and as a result, we might see greater movement of the forest floor litter layer on the more sloped plots relative to the upper slopes and ridges, which would lead to a thicker forest floor litter layer and a larger buffer against evaporative loss on plots that are up-slope. Further, lower plant and root biomass in a plot could reduce transpiration and thus increase soil moisture. And we do have fewer plants in these upper ridge sites. Finally, upper slopes and ridges have been found to have much deeper soils than those found on slopes, which could result in greater fluxes, especially if there are physicochemical changes that lead to greater aeration and thus more diffusion.

We are actively conducting research that will help prioritize these potential mechanisms. We have measured changes in microbial community composition, effects of warmer temperatures on fungal communities and fungal mats, effects of warmer temperatures on litter layer moisture and decomposition rates, have conducted soil incubation experiments, added deep soil pits along the topographic positions, and have analyzed how warming affects soil hydraulic properties. That these findings have spurred so much additional research and questions, further illustrates the value of the results for moving the field forward and generating greater understanding of how tropical forested ecosystems could respond to a warmer climate.

CO₂ fluxes were measured in very high temporal resolution (half hourly), but there is no deeper look into this high-resolution data. It would be interesting if the extremely high fluxes occurred randomly or if they were part of diurnal patterns or related to events such as rainfall.

We completely agree with the reviewer that deeper analysis of the high temporal resolution is of extreme interest. We did evaluate whether there were diurnal patterns in soil respiration and did not observe any. This is in line with a prior analysis conducted in neighboring forest that found no diurnal variation in soil respiration. In contrast, a moist forest site did show significant diurnal patterns with a mid-day depression in soil respiration values related to temperature. We have hypothesized that these different diurnal responses are due to the high rainfall environment compared with the drier forest site that experiences stomatal closure at mid-day. We additionally evaluated diurnal variation in available soil phosphorus and also found no significant variation for this forest, whereas diurnal patterns were observed for another site in Costa Rica. Combined, these data lead us to believe that temporal dynamics for this forest are driven by rainfall on a day-to-day basis rather than hourly time-scales. See citations for this work below:

- Wood, T. E., Detto, M., & Silver, W. L. (2013). Sensitivity of soil respiration to variability in soil moisture and temperature in a humid tropical forest. *PloS one*, 8(12), e80965.

- Gutiérrez del Arroyo, O., & Wood, T. E. (2020). Significant diel variation of soil respiration suggests aboveground and belowground controls in a tropical moist forest in Puerto Rico. *Journal of Geophysical Research: Biogeosciences*, 125(3), e2019JG005353.
- Wood, T. E., Matthews, D., Vandecar, K., & Lawrence, D. (2016). Short-term variability in labile soil phosphorus is positively related to soil moisture in a humid tropical forest in Puerto Rico. *Biogeochemistry*, 127, 35-43.
- Vandecar, K. L., Lawrence, D., Wood, T., Oberbauer, S. F., Das, R., Tully, K., & Schwendenmann, L. (2009). Biotic and abiotic controls on diurnal fluctuations in labile soil phosphorus of a wet tropical forest. *Ecology*, 90(9), 2547-2555.

We have now added a statement in the manuscript to clarify that there was no significant diurnal variation in soil respiration rates. See Line 130-131.

We also agree that developing a predictive model for assessing what drives this high temporal variability is of significant interest, including when hot moments occur. We have brought in new collaborators that are working on developing a model that will help us to better predict when these hot moments are likely to occur (i.e., Debjani Sihi at North Carolina State University, Eric Davidson at University of Maryland, and Jianqiu Zheng at Pacific Northwest National Laboratory), as part of a recently funded synthesis project. We are additionally including data collected as part of the Panama warming experiment to assist in this effort as part of this larger collaborative effort. Given the complexity of temporal dynamics and the randomness of the hot moments, the work we are doing to develop this predictive model is necessarily a separate manuscript and beyond the scope of this manuscript.

Authors thoroughly checked the instrumentation and tried everything to avoid measurement bias. This is now provided in the supplements (I only wonder that they did not observe high CO₂ fluxes when they “breathed directly into the instrument” – when I breath into my IRGA, the CO₂ concentration increases immediately to >10.000 ppm...). However, the most important information about chamber closure times and air flow, tubing length etc. during field measurements are nowhere to find in the ms.

Thank you for bringing this oversight to our attention. These data are now included as its own section of the supplementary methods where we describe the chamber closure time (2 minutes), cable lengths (15 – 30m) total tube volume, and data processing methods.

Soil moisture was an important explanatory variable of the temporal variability of the soil CO₂ efflux. The plot with extremely high CO₂ fluxes was permanently wetter than the corresponding control plot. For me there is no logical explanation that artificial understory and soil warming causes an increase in soil moisture. Typically, moisture decreases under warming or remains

similar if repeated rain wets the soil. Therefore, apparently, the much higher soil moisture in this particular plot was independent of the warming treatment – but might have had implications on the CO₂ efflux as shown in Suppl. Fig. 4. While control and warming plots at the two other treatment sites showed a similar response to moisture and temperature together, the control and warming plot at the upslope site (the one with extremely high emissions) showed a very different pattern. Control plot (5) soil CO₂ efflux was basically driven by soil temperature, while warmed plot (6) CO₂ efflux was driven by both temperature and moisture. This also suggests that soil moisture, which can be considered as independent of the soil warming treatment, had a huge effect on the high CO₂ fluxes from this plot. With a replication of only n=3 plot pairs, a single plot, which could have been influenced by other parameters as warming is quite critical.

As the reviewer outlines, we found that the upper ridge had higher soil moisture relative to the paired control, while the other pairs experienced slightly drier conditions. We believe that this difference is in large part what is so exciting and new about this work, and we see it as a strength of the manuscript because it helps us to tease apart direct and indirect effects of warming on soil respiration. As discussed above, we now provide multiple mechanisms through which warming could result in wetter soils, which we believe makes an important contribution to the field. For example, in a nearby forest, we evaluated temporal dynamics of soil respiration in response to a soil drying experiment and found that when soils are dry, moisture is the dominant control, and when soil moisture was high, temperature drove higher soil respiration rates. These findings are in line with that work. See Wood et al. 2013.

Wood, T. E., Detto, M., & Silver, W. L. (2013). Sensitivity of soil respiration to variability in soil moisture and temperature in a humid tropical forest. *PloS one*, 8(12), e80965.

In addition, there is a large body of research in this forest that demonstrates that a number of variables change with topographic position, which could contribute to higher soil moisture values in the upper slope plots. Ruling out all of the possible explanations is beyond what can be accomplished in this study, but based on the excellent reviewer suggestions, we do not provide a much richer look into the mechanisms that could explain these patterns. Elucidating these controls is something we continue to explore with additional research and that we believe will stimulate others to explore based on these results.

In the concluding paragraph L285-287 it is stated that: “Nevertheless, understanding the underlying mechanisms driving the response of soil carbon loss to warmer temperatures is critical...” - the current study does not provide these mechanisms (and a soil carbon loss was not detected either).

We have now changed this sentence to say “soil respiration” rather than “soil carbon loss” for clarity in Line 286.

There is gap of two whole months (Feb, Apr) in Fig 1 and 3 showing the soil CO₂ efflux. I could not find any explanation for this data-gap in the text. In L319 it is stated that equipment run almost continuously except during November.

Thank you for raising this issue. We have now provided more information on this gap in the methods. In early November there was a mechanical failure of several automated chambers

when some of the seals failed allowing water to enter during a period of especially high rainfall. We were not able to repair the chambers on site and sent them to LiCor for repair, which resulted in a gap in data collection. See lines 332-333.

The paired plots showed somewhat contrasting temporal response patterns to warming. Lower slope responded positively during the first months but responded already negatively to warming during the last months. Mid slope plot showed a negative response during some of the early phase but strong positive response later. Upslope plot also needed some time until it really increased dramatically. These contrasting temporal patterns could be discussed.

We whole heartedly agree that exploring the temporal dynamics and responses of soil respiration across these plots in response to warming is immensely interesting and we do plan to analyze these patterns in more depth. However, given the complexity and the high volume of data, we have engaged additional collaborators, including modelers, to help us conduct this work (see prior response above on new collaborations). These new analyses are beyond the scope and the space permitted for the results described in this paper, and thus we focus this paper on the treatment effects. We plan to dedicate a future manuscript to explore temporal dynamics.

The conclusion in Line 282 that “from warmed plots with no signs of declining in the first 1-2 years after initiating warming” is and overstated. Soil was warmed only during a single year. How to conclude on the second year?

We understand that this is confusing. In this statement we are referring to the combined results of both our study and that of the Panama experiment. Our study covered 1-year their study covered 2-years. In line 295 we now specify “In both sites...” prior to discussing the 1-2 years.

Reviewer #2 (Remarks to the Author):

In the original submission, reviewer 1 and myself raised concerns that (1) low replication meant that random spatial variation could explain apparent treatment differences, (2) analyses were pseudoreplicated and (3) the high soil respiration values were hard to explain and potentially implausible. The authors have gone to considerable effort to address these, with new field data and extensive new analysis. In relation to the original concerns:

1. The new field data on spatial variability helps to put this concern in context. While of course having more spatial replicates and greater spatial intermingling of treatments would be ideal, this new data does help (a) make the reader explicitly aware of the nature of the risk and (2) provide some reassurance that it doesn't drive patterns.

We are grateful for the past and present comments of this reviewer, the positive assessment of our work to address previous concerns, and we agree that the new data on spatial variability in soil respiration at the site help support the conclusion that the increase in soil respiration is due to the warming treatment rather than spatial heterogeneity. We think the

excellent comments of both reviewers substantially strengthened the dataset and manuscript.

2. The new GLS analysis is helpful and reassures me that differences aren't driven by treating all observations as independent.

We agree and deeply appreciated the suggestion that we revise our statistical approach.

3. I appreciate the extensive checks to make sure that the high readings were unlikely to be artefactual. Ultimately, I don't think we always need to be able to explain unexpected data as long as we are confident the data are real.

We fully agree that it is of utmost importance to have high confidence that our measurements are real, particularly with automated instruments. We are delighted that the reviewer is satisfied with the checks that went into evaluating the high readings.

In sum I think the original issues have been addressed or at least raised as transparent limitations.

I have two minor suggestions to consider:

- I am struck by the increased variability at higher temperatures, but discussion largely focuses on the mean. Perhaps a sentence or two on the implications of increased variability would be helpful.

Thank you for this suggestion. While the space we are allowed did not permit much addition of text, we do now include a statement about the higher variability in rates observed in the warmed plots. Please see Lines 76-78 and Lines 159-164. We do plan to include analyses of variability in a follow up paper focused more on temporal patterns, controls, and variability.

- To appreciate this variability, I would find it helpful in Fig. 2 to put all the graphs on the same y axis scale.

We have now included a version of Figure 2 in the Supplemental Figures (S5) where the values are on the same y axis scale. We kept the figures on different scale in the manuscript to illustrate the treatment effects, which are difficult to assess when all data are on the same axis given the substantial difference in scale.